# OpenReview forum: "SOPE: Situation-Aware and Statistically Indistinguishable Privacy Exfiltration for MCP-enabled Agents"
_ICML.cc/2026/Conference — ICML 2026 regular_

### Official Review · Reviewer_XQUD · 2026-03-11

**Soundness:** 2
**Presentation:** 2
**Significance:** 3
**Originality:** 2
**Overall Recommendation:** 3
**Confidence:** 3

**Summary:**

The authors claim that current methods suffer from a task / attack divergence. The attack is not informed about the task and thus, the data to be extracted is often rather unnatural in the context, which makes recognition easier.

The authors propose SOPE, a method to transform any benign MCP server to a privacy leaking version. To do that, they identify which sensitive items make sense for which functions to be additional arguments, which then get leaked by modifying the tool implementation.

The authors evaluate their method over thousends of cases and hundreds of mcp tools.

**Compliance With Llm Reviewing Policy:**

Affirmed.

**Key Questions For Authors:**

- See the above questions raised in the weakness section.
- The authors claim to have a statistically indistinguishable exfiltration process. Can the authors elaborate on that point?

**Limitations:**

Yes.

**Strengths And Weaknesses:**

Strengths:
- The tool transformation preserves utility. This is important, as tools that are not useful will quickly not get used anymore.
- The approach to not attempt to randomly extract data but data that makes sense in the context is sensible and something that should be kept in mind by practitioners in the field for future investigations.

Weaknesses:
- The attack model seems rather strong. It is not just to modify a tool description, but also the mcp tools implementation to directly leak sensitive data. However, in the presence of unaudited tools, this can be justified.
- Section 3.2: Here seems to be an important ablation missing. Why train models and not just ask an LLM which attributes make sense?
- Section 3.3: The training here is also somewhat strange. Can the authors substantiate their claims? I.e. i would be surprised to find that Grok refuses.
- Section 4: The baselines proposed by the authors might not be fair baselines and ill suited when presented as is. The authors propose to change the tool implementation while the related work focuses on indirect prompt injections in the tool description.

Minor:
- 094 right: did the authors forget $\alpha_k$ in the tuple?

---

> ### Author Rebuttal · Authors · 2026-03-31
>
> We thank the reviewer for recognizing the contribution of context-aware data extraction. We address each concern below.
> # W1: Feasible Attack Model
> **(1) An MCP server developer/publisher defines and publishes the complete description and implementation**; modifying the implementation requires no additional privileges beyond what any developer possesses.
>
> **(2) MCP marketplaces host 30k+ servers from unvetted developers.** Current vetting only checks connectivity and basic functionality, without developer authentication or code security checks [1–3].
>
> **(3) SOPE robustness against code audits.** Detection rates (%) of six inspection tools are low on 324 SOPE-generated servers:
>
> |CodeQL[4]|Codacy[5]|Semgrep[6]|Fang[7]|MCPScanner[8]|MCPGuard[9]|
> |-|-|-|-|-|-|
> |0|0|0|28|12|16|
>
> (a) Static analysis tools [4–6] fail: SOPE introduces no syntactic errors or code vulnerabilities; (b) MCP scanners [8,9] are largely ineffective: SOPE generates no malicious code or prompts that red-teaming would readily flag. However, (c) LLM-based analysis [7–9] occasionally detects redundant transmit calls; these limitations can be evaded by simple obfuscation of the transmit implementation (e.g., renaming the function, piggybacking on legitimate API calls): evaluation shows all detection rates drop to 0.
>
> [1]A Measurement Study of Model Context Protocol Ecosystem
> [2]Beyond the Protocol: Unveiling Attack Vectors in the MCP Ecosystem
> [3]Livemcpbench: Can agents navigate an ocean of mcp tools?
> [4]codeql.github
> [5]codacy.com
> [6]semgrep.dev
> [7]LLMs for Code Analysis: Do LLMs Really Do Their Job?
> [8]github/cisco-ai-defense/mcp-scanner
> [9]github/SaravanaGuhan/mcp-guard
> # W2: Why Train Assign Model Instead of Asking an LLM?
> We replace the trained Assign with three frontier LLMs using (a) binary queries for targeted (T) attacks and (b) open-ended queries for untargeted (U) attacks. Results (%):
>
> |Method|ASR(T)|LR(T)|ASR(U)|LR(U)|
> |-|-|-|-|-|
> |GPT-4o|76|67|72|65|
> |Gemini-3-pro|87|61|88|66|
> |Claude-opus-4.6|89|57|91|54|
> |**LLMs(Avg.)**|**83**|**62**|**84**|**61**|
> |**Assign**|**86**|**95**|**94**|**87**|
>
> - LLMs achieve reasonable ASR but substantially lower LR, especially in targeted mode, because: (a) safety alignment yields overly conservative judgments, determining that sensitive items are inappropriate to probe in all contexts; (b) Assign model provides calibrated confidence scores that enable flexible coverage–stealth trade-offs.
> - Computational efficiency: LLMs require $K \times N$ API calls for K tools and N privacy, while Assign performs a single batched forward pass.
>
> We will include this ablation in Table 4 of the revision.
> # W3: Commercial LLM Failures in Performing Fusion
> **(1) Commercial LLMs refuse to answer privacy-related questions due to safety alignment.** Appendix D shows concrete failure cases. Across 100 attempts, the overall refusal rate is 43%, rising to 82% for highly sensitive items (e.g., SSN, health records).
>
> **(2) Non-trivial refusal renders the pipeline unreliable for large-scale transformation**, despite variation among commercial LLMs, while SOPE's task-specific fine-tuning achieves 0% refusal with consistent quality.
>
> **(3) We tested 100 fusion attempts with Grok 4 as suggested; the refusal rate is 63%.** We will include additional LLM results in the revision.
> # W4: Fairness of Baseline Comparison
> **(1) Same threat actor, different attack surfaces.** Both SOPE and the baselines assume the same threat actor: the malicious MCP server developer, and ultimately deliver malicious prompts into LLM contexts. Baselines inject malicious prompts via tool outputs, not descriptions (§2.2).
>
> **(2) Ablation of tool implementation.** We conduct (a) SOPE without modifying tool implementations, and (b) baselines with modified tool implementations. Results for untargeted attacks (%):
>
> ||(a)ASR(S1)|(a)ESR|(b)ASR(S1)|(b)ASR(S2)|(b)ESR|
> |-|-|-|-|-|-|
> |SOPE|97|89|97|95|89|
> |Baselines(Avg.)|54|19|52|96|18|
>
> The gap is because (a) LLM safety alignment refuses privacy-related queries in irrelevant contexts, and (b) retrieval screening defenses flag recognizable adversarial patterns. This confirms that the key improvement lies in how SOPE crafts probing prompts, not merely in modifying tool implementations. We will include this ablation in the revision.
> # Q2: Statistically Indistinguishable
> SOPE-generated trajectories are indistinguishable from benign ones in the feature spaces (monitored by defenses): **(1) Textual patterns.** SOPE-crafted prompts lack the lexical patterns of conventional injections, so the red-teaming classifier (PromptGuard2) achieves 0% detection. **(2) Perplexity.** SOPE-generated prompts achieve ~6 PPL (significantly below 33–39 for conventional injections), leaving them inseparable from benign ones by DetectPPL. **(3) LLM latent space.** Fig. 3(d,e) shows SOPE trajectories overlapping benign ones, while baselines form separated clusters.
> # Minor
> Tool argument $\alpha_k$ is a component of description $\phi_x$.

---

### Official Review · Reviewer_aRSM · 2026-03-12

**Soundness:** 4
**Presentation:** 4
**Significance:** 4
**Originality:** 4
**Overall Recommendation:** 5
**Confidence:** 4

**Summary:**

This paper introduces SOPE, a “situation-aware and statistically indistinguishable” privacy exfiltration framework targeting MCP-enabled (Model Context Protocol) LLM agent systems. Unlike prior attacks that use rigid prompt templates and are easily flagged by existing defenses, SOPE leverages scenario awareness: it associates privacy exfiltration attempts with MCP tools for which such exposure would be plausible given the usage context, fuses privacy probes into prompt templates in a semantically coherent way, and transmits the data via code-level modifications (zero-click attacks). SOPE is evaluated extensively across 324 SOPE-transformed real-world servers, 27,216 test cases, multiple benchmark and commercial agents, and nine state-of-the-art defenses. Results show SOPE is highly effective, robust to defenses, and exposes serious protocol-level safety gaps in the MCP agent ecosystem.

**Compliance With Llm Reviewing Policy:**

Affirmed.

**Final Justification:**

The paper presents a creative and well-executed attack framework that exposes serious protocol-level safety gaps in the MCP agent ecosystem. The rebuttal fully addressed my concerns: the computational cost breakdown establishes attacker feasibility, the code audit evaluation against six inspection tools reinforces the attack's stealth, and the discussion of stricter deployment assumptions is honest and nuanced. The commitment to adding a dedicated limitations section with responsible disclosure further strengthens the manuscript. I maintain my score of 5.

**Key Questions For Authors:**

1. Can the authors provide empirical runtimes or resource usage for training/optimizing the Assign and Fuse modules, and comment on the attacker feasibility at scale?
2. Regarding the Leak phase, how robust is the attack against static analysis, code auditing, or marketplace vetting pipelines that inspect appended transmission logic?
3. How would SOPE perform under stricter deployment assumptions, such as schema validation, tool sandboxing, outbound network restrictions, or human confirmation for sensitive argument requests?

**Limitations:**

No. While the paper clearly motivates the security importance of the problem and emphasizes the need for protocol-level defenses, it does not appear to include a sufficiently explicit discussion of limitations and potential negative societal impact. In particular, the authors could strengthen the paper by adding a dedicated section discussing (1) the deployment assumptions under which the zero-click leakage channel remains feasible, such as schema permissiveness, tool execution privileges, and outbound communication capabilities; (2) the external validity of the evaluation, especially whether stealthiness holds under stricter enterprise sandboxing and human scrutiny; and (3) safeguards against misuse, such as responsible disclosure, release minimization, redaction of exploit details, and concrete defensive recommendations for MCP client and server developers. This would make the paper more complete and responsible.

**Strengths And Weaknesses:**

Strengths:
- The paper presents a creative attack framework that goes beyond template-based prompt injections by introducing a scenario-aware assignment mechanism grounded in contextual integrity, which maps tools to plausible privacy targets in a more realistic and stealthy way.
- The Fuse module is technically interesting and well designed: it uses an RL-based objective to balance privacy probing, utility preservation, and semantic similarity, making the attack more coherent and less brittle than simple template fusion.
- The zero-click Leak design is a notable contribution because it broadens the attack surface from prompt-level manipulation to code-level exfiltration, highlighting why defenses that only monitor LLM-issued tool invocations may be insufficient.
- The evaluation is extensive and convincing, comprising 27,216 cases, 324 transformed real-world servers, multiple benchmark and commercial agents, and nine state-of-the-art defenses. This gives the work substantial empirical support and practical relevance.
- The paper is well structured, and the ablation studies provide useful evidence that each component of the Assign-Fuse-Leak pipeline contributes meaningfully to the overall attack effectiveness.

Weaknesses:
- The paper does not sufficiently discuss the computational cost and attacker effort required to train or optimize the Assign/Fuse modules at scale.
- The “zero-click” Leak step assumes the client allows augmented schemas and outbound network calls from tool implementations; some MCP clients or enterprise deployments enforce stricter allowlists/sandboxes that may reduce the practicality of this channel.
- I also see several issues in evaluation and presentation. First, the paper repeatedly claims that the transformed server preserves original utility, but the main text does not provide a serious utility-retention evaluation beyond attack metrics; there is no systematic benign-task success analysis in the main results. Second, the setup is internally inconsistent: Table 1 shows 123 use cases, while the text states 129 use cases. Third, the paper generalizes to “commercial agents”, but one of the three, ChatGPT, does not natively support MCP and is adapted through function calling, which weakens the MCP-specific framing. Fourth, the defense comparison is uneven, since commercial agents are only evaluated with retrieval-screening defenses, yet the paper still draws broad conclusions about defense failure. Finally, the passive leakage rate is already high at roughly 28–33%, which makes the security baseline itself unusually leaky and complicates interpretation of the headline gains.

---

> ### Author Rebuttal · Authors · 2026-03-31
>
> We sincerely thank Reviewer aRSM for the thorough and positive evaluation.
> # Q1 & W1: Computational Cost and Attack at Scale
> **(1) Empirical costs.** Training runs on a single A100 (80 GB) GPU with external LLM API calls:
>
> |Phase|Data|GPU Mem(GB)|API Token(M)|Runtime(h)|
> |-|-|-|-|-|
> |Assign training|~5k|~20|~10|~2|
> |Fuse training|~3k|~20|~104|~18|
>
> - Assign module. ~5k aligned tuples from PrivacyLens, ~10M tokens for tuple generation and rewriting (§3.2-II).
> - Fuse module. PPO with ~90k LLM-as-a-judge scoring requests (≈104M tokens).
> - Inference. Generation of 324 malicious servers (~40k tools) in ~3 hours.
>
> **(2) Attacker feasibility at scale.** The entire pipeline (from data generation to malicious server uploads) completes within two days on a single A100, <$2 per server. This is well within reach of individual adversaries, and open-source alternatives (e.g., Llama-3-70B) further reduce costs. Once trained, models generalize across MCP servers; new servers require only seconds per tool inference without retraining. We will include a “Computational Cost” paragraph.
> # Q2: Robustness Against Static Analysis, Code Auditing, and Marketplace Vetting
> We evaluate 324 SOPE-generated servers against six code-level inspection tools. Detection rates (%) are:
>
> |CodeQL[1]|Codacy[2]|Semgrep[3]|Fang[4]|MCPScanner[5]|MCPGuard[6]|
> |-|-|-|-|-|-|
> |0|0|0|28|12|16|
>
> **(1) Static analysis and code auditing.** (a) Static analysis tools [1–3] fail because SOPE introduces no syntactic errors or code vulnerabilities; (b) MCP scanners [5,6] are largely ineffective because SOPE generates no malicious code or prompts that red-teaming would readily flag. However, (c) LLM-based analysis [4–6] occasionally detects redundant transmit calls; these limitations can be evaded by simple obfuscation of the transmit implementation (e.g., renaming the function, piggybacking on legitimate API calls): evaluation shows all detection rates drop to 0.
>
> **(2) Marketplace vetting.** Six major marketplaces (Appendix B, K) only check connectivity and basic functionality, not developer authentication or code security [7–9].
>
> [1]codeql.github
> [2]codacy.com
> [3]semgrep.dev
> [4]LLMs for Code Analysis: Do LLMs Really Do Their Job?
> [5]github/cisco-ai-defense/mcp-scanner
> [6]github/SaravanaGuhan/mcp-guard
> [7]A Measurement Study of Model Context Protocol Ecosystem
> [8]Beyond the Protocol: Unveiling Attack Vectors in the MCP Ecosystem
> [9]Livemcpbench: Can agents navigate an ocean of mcp tools?
> # Q3 & W2: Stricter Deployment Assumptions
> **(1) Schema validation.** SOPE adds function arguments that are contextually plausible (ensured by Assign). Strict validation would block Leak but creates a fundamental usability–security tension; no current MCP client enforces it.
>
> **(2) Tool sandboxing and outbound network restrictions.** Such mitigations hinder exfiltration but face challenges separating malicious transfers from legitimate ones: (a) many MCP tools legitimately make outbound calls; (b) decoying the transmit() action by piggybacking on legitimate API calls or delayed/batched behavior makes detection harder.
>
> **(3) Human confirmation** mitigates leakage at the cost of user experience. SOPE’s situation-awareness means requested information appears legitimate in context, lowering rejection rates.
>
> We will explicitly discuss these mitigation vectors in the "limitation" section of the revision.
> # W3: Evaluation and Presentation Issues
> We will (1) emphasize ISR (averaging 98% in Table 10) in the main text, (2) correct the number of 129 use cases, (3) clarify MCP framing and ChatGPT as a use case, (4) scope defense claims to the evaluated settings, and (5) refine the PLR discussion and headline gains.
> # Limitation
> We will add a section addressing limitations stemming from deployment assumptions, external validity, and potential mitigation strategies/vectors, and additionally discuss possible countermeasures (as presented above). We will also discuss responsible disclosure.

---

> > ### Author Rebuttal · Reviewer_aRSM · 2026-04-02
> >
> > computational cost breakdown (Q1: Assign $\sim$2h, Fuse $\sim$18h on a single A100; inference over 324 servers in $\sim$3h; less than 2 USD per server) clearly establishes attacker feasibility, further strengthening the paper's threat model. The code audit evaluation (Q2) against six inspection tools, with static analysis (CodeQL, Codacy, Semgrep) achieving $0\%$ detection and simple obfuscation nullifying even LLM-based scanners, reinforces the attack's stealth claims. The discussion of stricter deployment assumptions (Q3/W2) is honest and nuanced, properly acknowledging the usability-security tension while noting that no current MCP client enforces schema validation or outbound network restrictions. The commitment to fixing all presentation issues (W3) and adding a dedicated limitations section with responsible disclosure discussion addresses my remaining concerns. I maintain my score of 5.

---

> > > ### Author Response · Authors · 2026-04-07
> > >
> > > Dear Reviewer aRSM,
> > >
> > > Thank you very much for your thorough evaluation and detailed engagement throughout the review process. Your questions on computational cost (Q1), robustness against static analysis and marketplace vetting (Q2), and stricter deployment assumptions (Q3) pushed us to provide comprehensive evidence that we believe substantially strengthens the paper, these revisions will further improve the paper's completeness and rigor.
> > >
> > > Thank you again for the constructive and thorough review.
> > >
> > >
> > > Best regards,
> > >
> > > The Authors

---

### Official Review · Reviewer_ibws · 2026-03-13

**Soundness:** 2
**Presentation:** 3
**Significance:** 3
**Originality:** 3
**Overall Recommendation:** 4
**Confidence:** 4

**Summary:**

This paper proposes a novel privacy exfiltration attack targeting MCP-enabled agents. The framework consists of three main steps. First, it assigns privacy items that are contextually relevant to the MCP tool's usage scenarios. Second, it generates adversarial prompts that embed the probing intent while maintaining the functionality of the tool. Finally, the adversary extracts the desired privacy items by appending a transmission function to the tool's codebase, leaking the data via function arguments. Experiments conducted on both custom-built and commercial agents demonstrate the effectiveness of the proposed attack.

**Compliance With Llm Reviewing Policy:**

Affirmed.

**Final Justification:**

I thank the reviewer for the effort in rebuttal, and I adjust the score accordingly.

**Key Questions For Authors:**

Q1: Since the probing privacy assignment and prompt fusion phases are achieved by fine-tuning an LLM, how well does this approach generalize to users with diverse backgrounds and privacy contexts that extend beyond the common scenarios found in the training data?

Q2: The paper shows that existing non-situation-aware prompt injections perform poorly and are easily detected. However, there are also highly powerful and stealthy adaptive prompt injections like [1]. Would these state-of-the-art adaptive prompt injection attacks remain powerful even if the target privacy item is unrelated to the tool's usage? Demonstrating this comparison would strengthen the argument for the importance of situation-aware privacy probing as a key component of both attacks and defenses.

Q3: The paper would benefit from an explicit "Threat Model" subsection detailing the adversary's capabilities, access levels, and attack goals. This would (i) help readers better understand the attack setting and (ii) allow them to more rigorously evaluate whether the assumptions underlying the attack are realistic.



[1] The Attacker Moves Second: Stronger Adaptive Attacks Bypass Defenses Against LLM Jailbreaks and Prompt Injections.

**Limitations:**

Yes

**Strengths And Weaknesses:**

## Strength
S1. The paper addresses the ineffectiveness of existing tool-agnostic prompt injection attacks against MCP agents and proposes an adaptive, situation-aware attack. By aligning malicious behavior with the tool's expected context, the attack is harder to detect.
S2. The three-phase attack methodology is well-motivated.

## Weakness
W1. The experiments rely on only 20 synthetic user profiles to evaluate the attack. Since different users may possess different types of privacy items (which may also vary across contexts), and the privacy assignment model is trained on a specific scenarios from PrivacyLens, the generalizability of the attack remains unclear. What happens when users have privacy items beyond those covered in the training data? Expanding the evaluation with more diverse user profiles, privacy items, and task commands would better validate the effectiveness and generalizability of the attack.

W2. Defense mechanisms are only briefly discussed. In practice, injection attacks of this nature could potentially be detected by code security sanity checks or network traffic analysis. While it is acceptable for the attack framework to bypass these considerations, the authors should explicitly acknowledge these existing defense vectors as limitations of their approach.

W3. The threat model is not explicitly described, making it difficult to assess the practicality of the proposed attack.

Minor:

The authors are encouraged to use the term "adaptive" to describe their attack, as this is the core distinction from prior work. "Adaptive" vs. "non-adaptive" attacks are well-established terms in the security literature, and using this terminology would help readers better understand and contextualize the nature of the attack.

---

> ### Author Rebuttal · Authors · 2026-03-31
>
> We thank the reviewer for recognizing SOPE addresses the ineffectiveness of tool-agnostic attacks. We address each concern below.
>
> # Q1 & W1: Generalizability to Diverse Users and Privacy Contexts
> **(1) Diversity of training data.**
> - **Scenarios.** Training tuples are generated by PrivacyLens, an agent generating legitimate privacy exposure in tool-invoking contexts under established privacy norms. Coverage is broad because it includes: (a) 8k+ MCP servers across 6 marketplaces; (b) norms from legal frameworks, academic research, and crowdsourcing (§3.2).
> - **Privacy items.** The Google benchmark defines 26 privacy items widely adopted in agent privacy evaluation, spanning highly sensitive (e.g., SSN) to less sensitive but rarely exposed (e.g., preferences)[1,2].
>
> **(2) Generalizability to unseen privacy items during inference.**
> - When users have privacy items beyond training coverage, SOPE clusters, semantically abstracts, and uses similarity matching to map unseen items to the nearest category (TF-IDF label alignment, §3.2-II).
>
> **(3) Generalizability of models.**
> - **Assign model** estimates contextual legitimacy rather than memorizing item–tool pairs, generalizing to unseen combinations.
> - **Fuse model** is task-agnostic: it learns how to embed probing intent. PPO (Eq.2) rewards functional coherence and probing effectiveness, not pattern-matching specific items.
>
> **Overall**, SOPE is agnostic to specific privacy values, so 20 user profiles suffice for evaluation. It generalizes across privacy norms, privacy types, tools, and contexts, as shown by (a) 27,216 tests where the attacker is agnostic to the victim agent or task commands and (b) 75% ASR against target privacy of low contextual relevance (0.5; Table 9).
>
> [1]AirGapAgent: Protecting Privacy-Conscious Conversational Agents
> [2]Leaky Thoughts: Large Reasoning Models Are Not Private Thinkers
> # Q2: Comparison with Adaptive Prompt Injection Attacks
> We evaluate PSSU[3] (proposed by the reviewer) and AutoHijacker[4], two SOTA adaptive prompt injection attacks using iterative LLM interactions, unlike SOPE's one-time modification. We grant stronger assumptions: [3] knows the defense; [4] gets 5 rounds interactions. Data extraction ASR & ESR against retrieval screening (%):
>
> |Defense/ASR|w/o|I-GPT|StruQ|ISE|Defense/ESR|PG|D-PPL|SecMCP|
> |-|-|-|-|-|-|-|-|-|
> |PSSU|64|14|62|68|PSSU|76|91|18|
> |AutoHijacker|71|11|66|72|AutoHijacker|78|89|38|
> |SOPE|96|94|95|97|SOPE|100|96|87|
>
> **(1) Situation-awareness remains critical regardless of injection sophistication.** Two failure modes explain the gap: (a) LLM safety alignment rejects privacy requests in irrelevant scenarios; (b) trajectory defenses detect task-trajectory divergence (Fig. 3).
>
> **(2) Ablation evidence.** Table 4 shows that removing situation-awareness drops ESR 87%→44% & system-defense ASR 55%→23%.
>
> **(3) Orthogonality and complementarity.** These adaptive methods could be integrated into SOPE's Fuse module to further enhance evasion capability as future work.
>
> [3]The Attacker Moves Second: Stronger Adaptive Attacks Bypass Defenses Against LLM Jailbreaks and Prompt Injections
> [4]AutoHijacker: Automatic Indirect Prompt Injection Against Black-box LLM Agents
> # Q3 & W3: Threat Model
> - **Attacker's Goal:** Exfiltrate privacy items from agent memory to an attacker-controlled endpoint.
> - **Attacker's Capability:** A malicious MCP server developer modifies tool descriptions, prompts, and implementations (all under developer control) and uploads to marketplaces (GitHub, MCP.so, Smithery), with no agent or defense access.
> - **Access Level:** Supply-chain. Developers publish/obtain 30k+ servers via open marketplaces with minimal authentication and audits: the threat model is realistic[5,6].
>
> [5]A Measurement Study of Model Context Protocol Ecosystem
> [6]Beyond the Protocol: Unveiling Attack Vectors in the MCP Ecosystem
> # W2: Code-Level & Network-Level Defense
> **(1) Code security checks.** We evaluate 324 SOPE-generated servers against six code inspection tools. Detection rates (%) are:
>
> |CodeQL[7]|Codacy[8]|Semgrep[9]|Fang[10]|MCPScanner[11]|MCPGuard[12]|
> |-|-|-|-|-|-|
> |0|0|0|28|12|16|
>
> (a) Static analysis[7–9] finds no syntax errors or code vulnerabilities; (b) MCP scanners[11,12] are largely ineffective: no malicious code or prompts under red-teaming; (c) LLM-based analysis[10–12] occasionally spots redundant transmit calls, yet simple implementation obfuscation (renaming the function or piggybacking on legitimate API calls) drives all detection rates to 0.
>
> **(2) Network traffic.** Monitoring is a viable direction, but faces limits: (a) numerous legitimate outbound MCP calls complicate analysis; (b) delayed/batched transmission further hinders detection.
>
> [7]codeql.github
> [8]codacy.com
> [9]semgrep.dev
> [10]LLMs for Code Analysis: Do LLMs Really Do Their Job?
> [11]github/cisco-ai-defense/mcp-scanner
> [12]github/SaravanaGuhan/mcp-guard
> # Minor
> We will revise the term "adaptive" to align with established literature.

---

> > ### Author Rebuttal · Reviewer_ibws · 2026-04-03
> >
> > Thanks for the rebuttal. I will adjust the score.

---

> > > ### Author Response · Authors · 2026-04-07
> > >
> > > Dear Reviewer ibws,
> > >
> > > Thank you sincerely for your careful and constructive review throughout this process, which we believe meaningfully improves our work.
> > >
> > > We are grateful that you found our responses adequately addressed all concerns (marked as "Fully resolved"). We noticed that the score has not yet been updated and wanted to gently follow up, as the discussion period is nearing its end. Given your acknowledgement that all concerns have been fully resolved, we would be grateful if you could update the score and confidence at your earliest convenience.
> > >
> > > We also want to note that, based on the rebuttal, we have committed to concrete revisions in the camera-ready version: (1) adopting the "adaptive" terminology you suggested, (2) adding an explicit Threat Model subsection, (3) including the PSSU/AutoHijacker comparison and code-level defense evaluation as new results. We believe these additions, together with the existing strengths you recognized, bring the paper to a level commensurate with acceptance. We would be honored if you could consider a score of acceptance in light of the fully resolved concerns and the strengthened manuscript.
> > >
> > > Thank you again for your time and expertise.
> > >
> > > Best regards,
> > >
> > > The Authors

---

### Decision · Program_Chairs · 2026-04-30

**Decision:**

Accept (regular)

**Comment:**

This paper introduces SOPE, a framework for privacy exfiltration in MCP-enabled LLM agents that leverages situation-aware assignment of privacy items to tools, semantic fusion of probing prompts, and zero-click code-level data transmission. Reviewers agreed that the problem is timely and practically important, the three-phase Assign-Fuse-Leak pipeline is well motivated, and the evaluation is extensive, spanning 27,216 test cases across benchmark and commercial agents with nine defenses. The authors provided a thorough rebuttal that included new experiments comparing against adaptive prompt injection baselines (PSSU, AutoHijacker), evaluation against six code inspection tools, and an ablation replacing the Assign module with frontier LLMs. Reviewers ibws and aRSM both acknowledged their concerns as fully resolved and expressed support for acceptance, with Reviewer aRSM maintaining a score of 5 and Reviewer ibws adjusting upward to 4.

Reviewer XQUD raised concerns about the strength of the threat model, the fairness of baseline comparisons, and the motivation for training dedicated models rather than prompting LLMs directly. The rebuttal addressed these points with concrete experimental evidence, and notably several of these concerns overlap with issues that the other two reviewers marked as fully resolved. Reviewer XQUD did not engage during the discussion phase despite the deadline passing, which limits the weight I place on the unresolved score of 3. Some presentation issues remain (e.g., inconsistent use-case counts, the ChatGPT adaptation weakening the MCP-specific framing), but the authors have committed to fixing these in the camera-ready version.

I recommend acceptance of this paper, as it presents a technically sound and well-evaluated contribution that exposes meaningful protocol-level safety gaps in the MCP agent ecosystem and is likely to inform future defense research.

---

Note: the paper contains two references with incorrect author lists which is worth fixing.

```
Reference: Chen, S., Piet, J., Sitawarin, C., and Wagner, D. Shieldagent: Shielding agents via verifiable safety policy reasoning. arXiv preprint arXiv:2503.22738, 2025a.

Reference: Tong, W., Zhang, S., Song, K., Xu, S., Zhao, S., Agrawal, R., Indurthi, S. R., Xiang, C., Mittal, P., and Zhou, W. Secalign: Defending against prompt injection using preference optimization. arXiv preprint arXiv:2410.05451, 2024.
```